# Predictive Role of Serum Thyroglobulin after Surgery and before Radioactive Iodine Therapy in Patients with Thyroid Carcinoma

**DOI:** 10.3390/cancers15112976

**Published:** 2023-05-30

**Authors:** Alberto Signore, Chiara Lauri, Arianna Di Paolo, Valeria Stati, Giuliano Santolamazza, Gabriela Capriotti, Daniela Prosperi, Anna Tofani, Stefano Valabrega, Giuseppe Campagna

**Affiliations:** 1Nuclear Medicine Unit, Department of Medical-Surgical Sciences and of Translational Medicine, “Sapienza” University of Rome, 00185 Rome, Italy; 2Medical Oncology 2, IRCCS “Regina Elena” National Cancer Institute, 00144 Rome, Italy; 3Surgical Unit, Department of Medical-Surgical Sciences and of Translational Medicine, “Sapienza” University of Rome, 00185 Rome, Italy; 4Nuclear Medicine Unit, Sant’Andrea University Hospital, Via di Grottarossa 1035, 00189 Rome, Italy

**Keywords:** differentiated thyroid cancer, radioactive iodine, thyroglobulin

## Abstract

**Simple Summary:**

We measured serum thyroglobulin (Tg), a marker of thyroid cancer metastases, in patients operated on with total thyroidectomy for thyroid carcinoma on three different occasions at the beginning of follow-up: 30 days before radioactive iodine therapy (RAI) for remnant ablation (and at least 40 days after thyroidectomy in euthyroidism), at the time of RAI (in hypothyroidism), and 7 days after RAI in euthyroidism. Patients were then followed for at least 3 years searching for possible local recurrences or metastases. Results showed that the first Tg measurement is highly indicative and predictive of the future development of metastases.

**Abstract:**

Introduction: Thyroidectomy followed by radioactive iodine therapy (RAI) is the treatment of choice for differentiated thyroid carcinoma (DTC). Serum thyroglobulin (Tg) measurement has proved to be useful for predicting persistent and/or recurrent disease during follow-up of DTC patients. In our study, we evaluated the risk of disease recurrence in patients with papillary thyroid carcinoma (PTC), who were treated with thyroidectomy and RAI, by measuring serum Tg at different time-points: at least 40 days after surgery, in euthyroidism with TSH < 1.5 and usually 30 days before RAI (Tg_−30_), on the day of RAI (Tg_0_), and seven days after RAI (Tg_+7_). Methods: One hundred and twenty-nine patients with PTC were enrolled in this retrospective study. All patients were treated with ^131^I for thyroid remnant ablation. Disease relapse (nodal disease or distant disease) during at least 36 months follow-up was evaluated by serum measurements of Tg, TSH, AbTg at different time points and by imaging techniques (neck ultrasonography, ^131^I-whole body scan (WBS) after Thyrogen^®^ stimulation). Typically, patients were assessed at 3, 6, 12, 18, 24, and 36 months after RAI. We classified patients in five groups: (i) those who developed nodal disease (ND), (ii) those who developed distant disease (DD), (iii) those with biochemical indeterminate response and minimal residual thyroid tissue (R), (iv) those with no evidence of structural or biochemical disease + intermediate ATA risk (NED-I), and (v) those with no evidence of structural or biochemical disease + low ATA risk (NED-L). ROC curves for Tg were generated to find potential discriminating cutoffs of Tg values in all patients’ groups. Results: A total of 15 out of 129 patients (11.63%) developed nodal disease and 5 (3.88%) distant metastases, during the follow-up. We found that Tg_−30_ (with suppressed TSH) has the same sensitivity and specificity than Tg_0_ (with stimulated TSH), and it is slightly better than Tg_+7_, which can be influenced by the size of the residual thyroid tissue. Conclusion: Serum Tg_−30_ value, measured in euthyroidism 30 days before RAI, is a reliable prognostic factor to predict future nodal or distant disease, thus allowing to plan the most appropriate therapy and follow-up.

## 1. Introduction

Currently, surgery, radioactive iodine therapy (RAI), and hormonal suppression are the treatments of choice for patients with differentiated thyroid cancer (DTC). Despite the effectiveness of these treatments, DTC prognosis is influenced by recurrence of disease, which occurs in up to 30% of patients, depending on initial therapy and other prognostic factors.

Most of these recurrences (66%) occur within the first 2–3 years after surgery [1]. Mazzaferri et al., in 1994, demonstrated that local recurrences are frequent, especially in the cervical lymph nodes (74%), followed by thyroid remnants (20%), and then trachea or muscle (6%). Only 8% of patients with local recurrences died of cancer. By contrast, 21% of DTC patients develop distant recurrences, most often represented by lung metastases. In this case, the percentage of patients who died from cancer is approximately 50% [2].

In the past, the standard approach to DTC involved total thyroidectomy followed by RAI ablation/therapy and long term TSH suppression with levothyroxine administration, without regard to recurrence risk.

Nevertheless, there is a vivid debate about how to manage these patients, whether to perform RAI ablation of residual thyroid tissue or not. It is, therefore, crucial to identify appropriate risk stratification systems and specific tumor markers with a high prognostic value to guide “tailored to patient” decision-making [3].

The role of serum thyroglobulin (Tg) as a tool to monitor disease status during follow-up is well established. Being produced by normal thyrocytes and by differentiated malignant cells, it is particularly helpful, during the follow-up of patients with DCT after ablative therapy (post-ablation Tg), to predict the presence of persistent or recurrent disease after initial therapy [4]. Undetectable levels of serum post-ablation Tg, with or without TSH stimulation, indicate a stable remission of the disease, while an increasing concentration of serum post-ablation Tg suggests the presence of loco-regional or distant disease. Moreover, the availability of more sensitive Tg assays, such as the second- or third-generation immunometric assays with a high sensitivity (≤0.10 ng/mL), in many cases obviates the need for rh-TSH stimulated Tg tests and makes serum Tg measurement a quick and cost-effective tool in DTC management. Nevertheless, the presence of Tg antibodies (TgAb) may underestimate Tg levels, thus representing a possible limitation to the ultrasensitive Tg measurement [5]. Since TgAb concentration varies according to Tg levels, and thereby to changes in thyroid tissue mass, TgAb represents a surrogate of Tg and can be used as a tumor marker during follow-up [5,6].

Despite the role of serum Tg before RAI having not been extensively elucidated, an optimal assessment of post-operative disease status may suggest additional treatment (RAI ablation/therapy, additional surgery, or other treatment). Therefore, there is an arising interest in evaluating its prognostic role in the postoperative risk stratification (after surgery and before RAI ablation/therapy). Given a half-life of 65 h, the postoperative Tg reaches its nadir by 3–4 weeks after surgery, thus a valid time point to measure it can be after at least 40 days form surgery on levothyroxine therapy. Moreover, it can be helpful to investigate if post-ablation Tg, tested 7 days after ^131^I administration, could be predictive of the disease status.

## 2. Study Endpoints

The primary endpoint of our study was to evaluate the role of Tg measured at multiple time points in predicting persistence and/or recurrence of local and distant disease: in euthyroidism, with TSH ≤1.40 mIU/L, at least 40 days after surgery, and approximately 30 days before RAI remnant ablation/therapy (Tg_−30_); Tg measured at time of RAI remnant ablation/therapy after TSH-stimulation (Tg_0_) induced by hypothyroidism (Tg_0H_) or rhTSH administration (Tg_0TSH_); and Tg seven days after RAI remnant ablation/therapy (Tg_+7_) obtained after inducing hypothyroidism (Tg_+7H_) or administering rh-TSH (Tg_+7TSH_).

The secondary endpoint was to identify a consistent cutoff for Tg values at each time point, to reliably discriminate patients with persistent/recurrent disease from disease-free patients.

## 3. Patients and Methods

### 3.1. Study Design and Patient Selection

In this retrospective, mono-institutional, observational study, we analyzed medical record data of all patients who underwent total thyroidectomy for DTC, with or without neck nodes dissection, between 2013 and 2016, and received RAI in the Nuclear Medicine Unit of S. Andrea Hospital of Rome, Italy. We generated a database, using the Microsoft Excel^®^ program, among patients with DTC, and in particular with papillary thyroid carcinoma (PTC). The database included data from unstructured documents (i.e., medical notes) and structured data (i.e., laboratory results, treatment information, etc.). In particular, we collected demographic data; medical history; thyroid and neck US scans; serum levels of TSH, Tg, TgAb, and levothyroxine therapy; surgical reports and histopathology; and RAI therapy and radiological findings and outcomes.

The date of the first diagnosis of PTC was considered the date of surgery.

The inclusion criteria were patients ≥ 18 years old; patients who underwent a total/near total thyroidectomy +/− neck nodes dissection within 12 weeks prior to RAI for thyroid remnant therapy; papillary histology (including papillary carcinoma follicular subtype); data on Tg, TSH, and AbTg from the same laboratory; follow-up of at least 36 months. All patients had a postoperative neck ultrasound, with multifrequency probes (7.5–12 MHz) and color Doppler, to exclude lymph node metastasis and to better plan the subsequent therapeutic procedures. All patients were treated with levothyroxine substitution therapy at suppressive (with TSH between 0.1 and 0.3 mIU/L) or non-suppressive doses (with TSH < 1.5 mIU/L), based on the ATA risk assessment and clinical evaluation by the physicians.

The exclusion criteria were aggressive histological PTC variants (i.e., >10% tall cell, columnar, hobnail, and solid variants); positive TgAb at any time during the follow-up period; other malignant neoplasms or Hashimoto’s thyroiditis (because of high TgAb titre); any iodinated contrast agents within 3 months before the RAI; patients who took, within 4 weeks prior to administration, drugs or foods that affect iodine uptake or metabolism, such as multivitamins, glucocorticoids, diuretics, lithium, thiouracil, tazobactam, algae, or iodine (except thyroid hormone replacement therapy); pregnant or lactating women; and patients enrolled in any clinical trials.

Follow-up was defined as the time between RAI ablation/therapy and the last documented laboratory tests (TSH, Tg, and TgAb) and/or physical examination and/or imaging studies.

### 3.2. Surgical Approach and Diagnostic Workup before RAI

All patients underwent total thyroidectomy and therapeutic central-compartment (level VI) ± lateral (levels II–V) selective neck dissection, if clinically proven nodal metastasis; prophylactic central-compartment (level VI) was performed according to American Thyroid Association (ATA) Guidelines [3,7]. Histology was evaluated only by our center pathologists.

Disease stage (TNM) was determined according to the classification system of the American Joint Committee on Cancer (AJCC-TNM staging 2010) [8], and the postoperative risk assessment was defined according to ATA recommendations [3].

### 3.3. RAI Ablation/Therapy

The decision for RAI therapy was based on clinicopathologic factors, such as tumor size, histology, nodal metastasis, extra-thyroidal extent, residual disease and distant metastasis, and patient’s age. The decision on administered therapeutic RAI activity was based on postoperative risk stratification of the patient as defined by the ATA guidelines [3,7].

RAI ablation/therapy was performed inducing a hypothyroid state or after rh-TSH (Thyrogen, Genzyme, Cambridge, MA, USA) administration with the suggested published protocol [9]. A serum a TSH concentration of >30 mIU/L was considered adequate to validate the administration of ^131^I for ablative/therapeutic purpose. The reason for choosing one or the other protocol mainly depended on the availability of Thyrogen, regardless of patient age or ATA risk.

All patients observed a 10-day low iodine diet before RAI ablation/therapy and were hospitalized in shielded rooms following adequate radiation protection standards. They were discharged on levothyroxine suppressive therapy with indications on the radiation protection rules to be followed.

After 7 days from RAI treatment, the patients were evaluated with a post-therapy ^131^I whole-body scan (WBS). WBS was performed with a gamma camera with a large field of view, a high-energy collimator, and a 2.5 cm thick crystal (Philips, Forte gamma camera), according with the current EANM guidelines [10].

The patient lay supine with a moderate head reclination; anterior and posterior static images of the whole body were recorded, taking at least 20–30 min per view. Ten minutes static anterior and posterior images of neck and mediastinum were also obtained with the same gamma camera, with a large field of view, a high-energy collimator, and a 256 × 256 matrix. Images were interpreted qualitatively by visual assessment of the size and tracer uptake intensity of the residual tissue or distant metastases.

### 3.4. Thyroglobulin, Anti-Thyroglobulin Antibodies and Thyrotropin Measurements

Blood samples were collected to measure serum levels of Tg, TgAb, and TSH at different time points: in the euthyroid state at least 40 days after surgery and approximately 30 days before ablation therapy (Tg, TgAb, and TSH_−30_); at the same day of RAI ablation/therapy just before ^131^I administration after TSH-stimulation induced by hypothyroidism (Tg, TgAb, and Tg_0H_) or rhTSH administration (Tg, TgAb, and Tg_0TSH_); 7 days after RAI ablation/therapy obtained after inducing hypothyroidism (Tg, TgAb, and Tg_+7H_) or administering rhTSH ((Tg, TgAb, and Tg_+7TSH_); during follow-up at 3, 6, 12, 18, and 24 months. TSH, Tg, and TgAb levels were measured in the same laboratory by using the same assay (Abbott^®^ Architect I 1000SR platform, Princeton, NJ, USA) and the same ultrasensitive second-generation kits (CMIA, Architect, Abbott^®^, Princeton, NJ, USA).

### 3.5. Imaging during Follow-Up

During follow-up, a second diagnostic WBS was performed at 12 or 18 or 24 or 36 months after RAI ablation/therapy in case of suspicion of metastases, by the administration of 185 MBq (5 mCi) of ^131^I per os, and after two days, an injection of rh-TSH. Diagnostic WBS was performed 24 h after the ^131^I administration (in the fourth day), while Tg, TSH, AbTg measurements were performed on the first day before the first injection of rh-TSH (basal measurement) and on the third and fourth day (stimulated measurements).

For the follow-up WBS, we used the same gamma camera, execution modality, and interpretation criteria of the post-therapy scan. The WBS was considered positive if there was clear evidence of malignant ^131^I-positive lesions. Minimal thyroid remnants were considered negative for structural evidence of disease.

Neck ultrasound was performed in all patients at 3, 6, 12, 18, 24, and 36 months of follow-up, by the same expert operator. The ultrasonographic criteria for malignancy were the following: the presence of lesions in the thyroid bed displaying hypo-echogenicity; neck lymph nodes without hilum and a rounded shape and the presence of increased vascularization, microcalcifications, irregular margins, and being taller-than-wide in the transverse plane in the thyroid bed.

During follow-up, further diagnostic imaging modalities, i.e., radiography, CT, neck MRI, and [^18^F]FDG PET/CT, were performed in patients with high Tg levels and negative WBS to detect possible local or distant metastases.

The follow-up program adopted in the study is summarized in Figure 1.

Disease-free status was defined, according to ATA guidelines, as the presence of all the following: no clinical evidence of residual tumor; an absence of pathological functioning thyroid tissue in the thyroid bed and/or of distant metastases on RAI imaging on ultrasound scan; Tg values < 0.2 ng/mL (on TSH suppression); or a stimulated Tg level < 1 ng/mL in the absence of interfering antibodies [3,7].

In case of recurrent disease (local or distant metastases), surgery, external radiotherapy, RAI therapy, and/or tyrosine-kinase inhibitors were considered.

### 3.6. Patient’s Classification

After initial therapy and during follow-up, we evaluated both structural and biochemical persistence or recurrence of disease. Biochemical recurrence of disease was considered, such as an abnormal Tg findings or rising of AbTg levels during follow-up. Presence of disease was defined as presence of persistent or recurrent, locoregional, or distant structural pathological findings. Combining the presence/absence of disease after initial therapy (surgery and RAI ablation/therapy) and during follow-up and ATA initial risk, we obtained the following groups:(i)Patients with evidence of locoregional structural disease (nodal disease, ND);(ii)Patients with evidence of distant structural disease (distant disease, DD);(iii)Patients with no evidence of structural disease but with a biochemical indeterminate response and minimal residual thyroid tissue after RAI ablation/therapy (R);(iv)Patients with no evidence of structural or biochemical disease and intermediate ATA risk (NED-I);(v)Patients with no evidence of structural or biochemical disease and low ATA risk (NED-L).

### 3.7. Ethical Requirements

The study was conducted in accordance with the guidelines for the reporting of tumor marker studies (REMARK) [11].

Due to the retrospective nature of this study, the local ethical committee was only notified.

### 3.8. Statistical Analysis

Continuous variables in Table 1 are shown as mean ± SD (standard deviation) and range, while categorical variables are presented as absolute frequencies and percentage—*n* (%).

The normality/symmetry of the continuous variables and of the residuals was evaluated by Shapiro–Wilk test and checking Q–Q plot.

The Kruskal–Wallis test was used to verify differences among groups of patients (nodal/distant disease, biochemical indeterminate response + minimal residual, or no evidence of structural disease + low/intermediate ATA risk), and the continuous variables analyzed and the data are shown as median and range. The homoscedasticity was verified by checking studentized residuals vs. fitted values plot. Post hoc analysis was performed by Dwass–Steel–Critchlow–Fligner tests in order to correct the multiple comparisons. A Mann–Whitney test was used to analyze differences between levothyroxine withdrawal vs. rh-TSH patients at RAI therapy relative to the continuous variables.

Youden’s index, minimum distance, and minimum difference were used in conjunction with receiver operating characteristic (ROC) to determine the optimal cutoff value to predict recurrence of disease, comparing nodal disease vs. non-evidence of disease and distant disease vs. non-evidence of disease.

In the presence of small samples, few and rare events of complete separation lead to the non-convergence of maximum likelihood (ML) logistic regression estimates; therefore to mitigate the bias caused as a consequence of the insufficiency of events, Firth’s penalized likelihood [12] was applied. This technique was a solution used to minimize the analytical bias caused by complete separation, small samples, and rare events.

Cutoff values were chosen when at least two out of three methods provided the same result. When the three methods were different, Youden’s index was chosen to identify the threshold.

All analyses were performed by SAS vs. 9.4 and JMP PRO vs. 16 (SAS Institute Inc., Cary, NC, USA). A *p* < 0.05 was considered statistically detectable.

## 4. Results

Between 2013 and 2018, 4025 consecutive patients with PTC underwent total or near-total thyroidectomy in our institution. Among these, we excluded 615 patients with incomplete data or incomplete follow-up and a total of 345 patients, which matched the exclusion criteria, as reported in Figure 2. Therefore, 129 patients (30 males and 99 females; mean age: 57.86 ± 13.24 years) with complete follow-up and verified clinical and serological data were included in this study.

Total thyroidectomy followed by radioiodine remnant ablation was performed in all enrolled patients. Central neck lymph node dissection was performed in 28 (21.71%) cases and lateral + central compartment neck dissection in 18 (13.95%) patients.

A classic papillary variant of thyroid cancer was found in 59 patients (45.74%), a follicular variant in 47 (36.43%), a mixed papillary cancer in 13 (10.08%), a papillary variant with less than 10% tall cells in 5 (3.88%), a papillary oncocytic variant (oxyphilic or Hurtle cells) in 4 (3.10%), and a sclerosing variant in only one patient (0.78%) (Table 1).

All patients had no evidence of metastases at the time of RAI.

RAI remnant ablation/therapy was performed after 4 weeks of levothyroxine withdrawal in 54 (41.86%) patients and after two days of rh-TSH i.m. injection in 75 (58.14%) patients.

The mean ^131^I activity was 3404.74 ± 1094.83 MBq (range: 1110 to 5550 MBq), even though we preferably used a dose of 1850 MBq, as previously described [13].

Table 2 shows the patients’ distribution among the five groups according to ATA risk.

A comparison between administered ^131^I activity, median Tg, and TSH levels at different time points in the five groups of patients is shown in Table 3.

From post hoc analysis, differences were found in median Tg_−30_ levels among the ND group and DD and between NED-L and NED-I; while the DD group median Tg_−30_ differed from R, NED-I, and NED-L. Figure 3 shows the comparison of median Tg values in ND patients versus both NED-I and NED-L grouped together.

Given the known influence of TSH on Tg values, we also performed the analysis for TSH, demonstrating that there was no difference in TSH_−30_ levels among all these five groups of patients; thus, the different values of TSH did not interfere with Tg_−30_ in our population_._ Additionally, Tg_0_ median values were different among the groups and were not influenced by patient preparation (i.e., levothyroxine withdrawal versus rh-TSH stimulation), except for the R group.

No difference in TSH_0_ levels among all these five groups of patients was found. For Tg_+7_, different median values were observed among the ND group versus NED-L and NED-I group and by DD vs. the NED-I group. No interference of patient preparation (i.e., levothyroxine withdrawal versus rh-TDH stimulation) was demonstrated.

A difference in ^131^I-administered activity from the NED-L group and, respectively, the nodal disease and the distant disease groups was consistent with the fact that NED-L groups patients with lower overall risk, which motivates a less aggressive management in terms of ^131^I activity.

Tg cutoff values have been identified in multiple time points for ND versus NED (both NED-L and NED-I) and DD versus NED using Youden’s index, minimum distance, and minimum difference in conjunction with the receiver operating characteristic (ROC) method.

For ND versus NED (L+I), the cutoff of Tg_−30_ was 1.30 ng/mL with sensitivity (Se) = 100 and specificity (Sp) = 89.0. For DD versus NED (L+I), the cutoff of Tg_−30_ was 16.00 ng/mL with Se = 100.0 and Sp = 100.0 (Table 4, Appendix A).

## 5. Discussion

The results of the present study provide pertinent data about the recurrence of disease in PTC patients treated with thyroidectomy and RAI, which are commonly performed in PTC patients [14,15,16].

There are many controversies with regard to the initial treatment and follow-up protocols in PTC patients, especially for low-risk DTC [17,18,19]. One of the most important matters of debate is the activity to administer. Several studies supported that remnant ablation with a low activity of ^131^I is as effective as ablation with higher activities, does not induce chromosomal damage, and is not associated with reduced disease-free survival compared to the use of higher administered doses [13,20,21,22].

In our study, the administered ^131^I activity was chosen depending on the ATA risk, and only 6.1% of enrolled patients had a disease recurrence. Interestingly, among patients who developed distant metastases, one was treated with 100 mCi, one with 120 mCi, and three with 150 mCi. Among patients who developed nodal metastases, one was treated with 30 mCi, one with 50 mCi, three with 100 mCi, seven with 120 mCi, and three with 150 mCi. Therefore, most patients developed nodal or distant metastases despite being treated with high ^131^I doses. We may argue whether, in these patients, it would have been better to administer a lower dose of ^131^I in view of possible further treatment with ^131^I, thus sparing irradiating too much the bone marrow.

Another critical issue is the role of serum markers to guide the most tailored treatment and a personalized follow-up. Several studies confirmed an increased risk of persistent/recurrent disease after initial therapy (surgery and RAI ablation/therapy) in patients with post-operative TSH-stimulated Tg > 1–2 ng/mL at the time of ablation [23,24,25], with an optimal balance of sensitivity and specificity for thyroid hormone withdrawal in postoperative Tg values between 20 and 30 ng/mL [26]. Postoperative stimulated Tg values (>10–30 ng/mL) are also associated with poor survival [27].

However, the value of Tg as a predictive factor before ^131^I treatment is controversial due to the existence of residual thyroid tissue after surgery [28,29]. Tg levels may be influenced by multiple factors (i.e., TSH level, the mass of residual malignant and/or normal thyroid tissue, the individual risk of having loco-regional or distant metastasis, and/or the sensitivity of the Tg assay) [30,31].

Moreover, the correct timing to measure post-operative TG and the optimal cutoff for stimulated or suppressed postoperative Tg levels, which can confirm or exclude the presence of lesions, have not been identified yet [32,33,34].

Therefore, the primary outcome of our study was to evaluate the prognostic value of suppressed and TSH-stimulated Tg at multiple time points (Tg_−30_, Tg_0_, and Tg_+7_) to clarify its predictive role for future recurrence in PTC patients.

The hypothesis to measure Tg_−30_ was that if nodal or distant metastases are present at the time of surgery, Tg levels in euthyroidism could be higher than in patients without metastases. By contrast, the hypothesis to measure Tg_+7_ was that if nodal or distant metastases are present at the time of RAI, Tg levels could be higher than in patients without metastases due to tissue necrosis induced by ^131^I and subsequent Tg release. In both cases, we wanted to investigate if metastases can be supposed despite the presence of a thyroid tissue remnant that produces Tg itself.

In our study, we included only post-operative patients without AbTg and with a TSH level below 1.5 mIU/L, and Tg values were measured by an ultrasensitive test. This strict selection could be a limitation of the study but, indeed, allowed us to define a subgroup of patients in which a Tg measurement, as early as 40 days post-surgery, with thyroid remnant, can be used as a reliable prognostic tool to correctly manage the patient. Indeed, we identified a cutoff for post-operative Tg value’s ability to reliably distinguish patients with nodal disease or distant metastases from disease-free patients. Unexpectedly, Tg_−30_ was more sensitive and specific than Tg_0_ and Tg_+7_, regardless of whether patients were treated with ^131^I in hypothyroidism or with stimulation.

Other important findings that deserve to be discussed are that patients who developed nodal metastases were classified in five cases as low ATA risk, in nine cases as intermediate ATA risk, and only in one case as high ATA risk (Table 2). Patients that developed distant metastases were initially classified as intermediate ATA risk in four cases and as high ATA risk only in one case (Table 2). It emerges that patients with low ATA risk should also be followed up carefully, particularly if Tg levels are above the defined threshold.

The major strength of this study is the central assessment of serum Tg, TSH, and TgAb measurements and imaging data (neck ultrasound and WBS) during 36 months of follow-up, which warrants the comparability of results at each time point. This has been possible thanks the creation of a therapeutic diagnostic pathway (PDTA) for patients with DTC.

Moreover, our analysis confirmed that the percentage of recurrent disease could differ according to the different follow-up protocols adopted. Local or distant recurrences of metastases (*n* = 20) were diagnosed when radioiodine WBS was added to the follow-up protocol at 12 or 24 months.

Despite the interesting findings, this study has some limitations. First of all, the short follow-up, as we know that nodal metastases may become evident up to 20 years after surgery. Another limit is related to the low sample size, although we deliberately wanted to investigate a selected population. Finally, it is a retrospective and monocentric study that needs to be confirmed by larger multicenter studies.

## 6. Conclusions

With this retrospective study, in a selected population of patients operated on for PTC, we showed that serum Tg, measured with highly sensitive assays 40 days after surgery in euthyroidism, is a reliable prognostic marker that should be used to guide patient follow-up. In patients with Tg < 1.3 ng/mL, RAI could be avoided or administered with low doses of ^131^I. If Tg levels are between 1.3 and 16 ng/mL, more investigations should be carried out to search for nodal metastases (ultrasound, PET/CT, etc.). Finally, Tg levels higher than 16 ng/mL should alert the clinician about possible distant metastases.

## Figures and Tables

**Figure 1 cancers-15-02976-f001:**
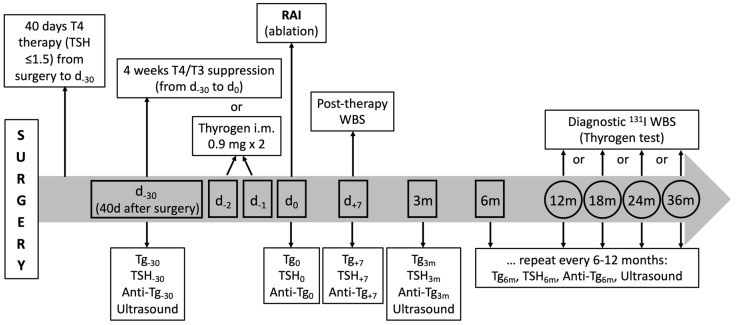
Schematic representation of biochemical and imaging tests performed in patients with PTC after surgery during the 36-month follow-up. The graph also shows when Tg, AbTg, and TSH are measured.

**Figure 2 cancers-15-02976-f002:**
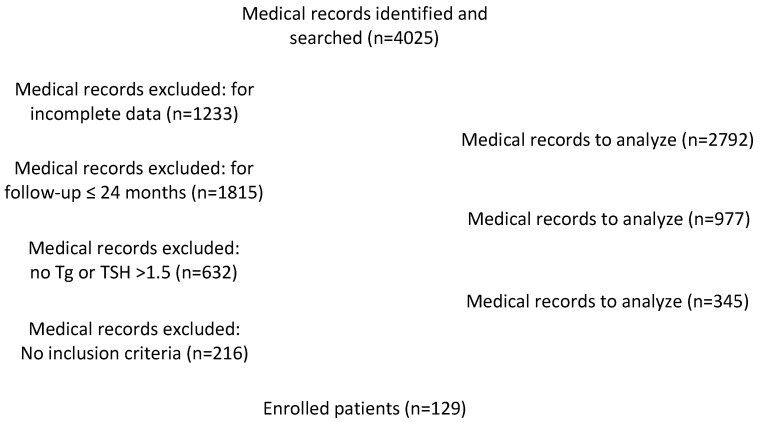
Flow chart for medical records identification.

**Figure 3 cancers-15-02976-f003:**
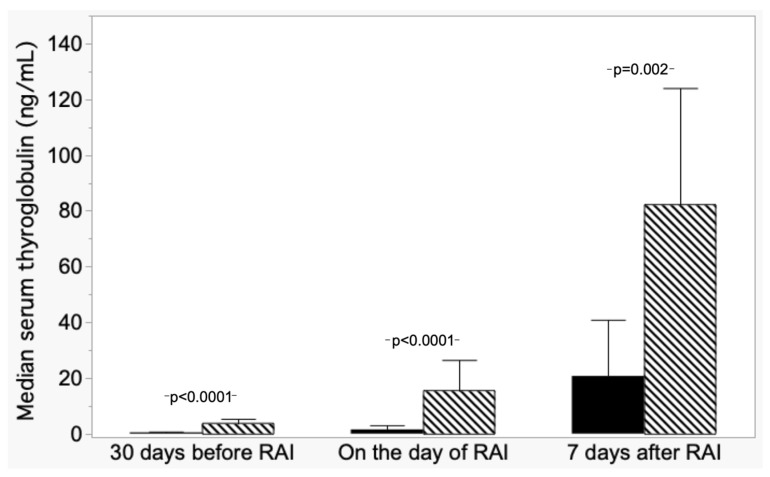
Trend of median serum thyroglobulin (±absolute deviation) in patients with nodal disease (ND, hatched columns) and patients with non-evidence disease (NED L+I, dark columns).

**Table 1 cancers-15-02976-t001:** Clinical characteristics of enrolled patients.

Parameter	Values
Age at diagnosis, mean ± SD, (range) (years)	51.21 ± 13.77, (21 to 84)
Age (years)	
Male, mean ± SD, (range)	62.24 ± 12.62, (29.33 to 84.71)
Female, (mean ± SD), (range)	56.62 ± 13.06, (28.86 to 90.07)
Sex, n (%)	
Male	30 (23.26)
Female	99 (76.74)
Histology, n (%)	
Classic papillary	59 (45.74)
Papillary follicular variant	47 (36.43)
Papillary mixed variant	13 (10.08)
Papillary tall cells variant	5 (3.88)
Papillary oncocystic variant	4 (3.10)
Papillary sclerosis variant	1 (0.78)
Multifocality, n (%)	
Yes	52 (40.31)
No	77 (59.69)
Capsular invasion, n (%)	
Yes	41 (31.78)
No	7 (5.43)
Not available	81 (62.79)
Vascular invasion, n (%)	
Yes	10 (7.75)
No	109 (84.50)
Not available	10 (7.75)
Margins, n (%)	
R0	108 (83.72)
R1	12 (9.30)
Not available	9 (6.98)
AJCC TNM classification, n (%)	
T1	60 (46.51)
T2	15 (11.63)
T3	53 (41.08)
T4	1 (0.78)
Nx	77 (59.69)
N0	21 (16.28)
N1	31 (24.03)
ATA risk, n (%)	
Low	78 (60.47)
Intermediate	49 (37.98)
High	2 (1.55)
Patient preparation, n (%)	
rhTSH	75 (58.14)
4 weeks Levotyroxine withdrawal	54 (41.86)
RAI dose, n (%)	
1850 MBq (50 mCi)	34 (26.36)
1850–3700 MBq (50–100 mCi)	22 (17.05)
>3700 MBq (>100 mCi)	73 (56.59)

Abbreviations: SD—standard deviation, R—margins, AJCC—American Joint Committee on Cancer, T—primary tumor, N—lymph node status, M—distant metastasis, ATA—American Thyroid Association, RAI—radiodine therapy, rhTSH—recombinant human TSH. Staging according to the AJCC TNM classification system for differentiated thyroid cancer carcinoma (7th edition).

**Table 2 cancers-15-02976-t002:** ATA risk for different groups of study.

	ATA Risk			
	*n* (%)	Low *n* (%)	Intermediate *n* (%)	High *n* (%)
Nodal Disease (ND)	15 (11.63)	5 (33.33)	9 (60.00)	1 (6.67)
Distant Disease (DD)	5 (3.88) *	0 (0.00)	4 (80.00)	1 (20.00)
Biochemical Indeterminate—Minimal residual tissue (R)	8 (6.20)	2 (40.00)	6 (60.00)	0 (0.00)
Non-evidence of disease—ATA intermediate risk (NED-I)	30 (23.26)	0 (0.00)	30 (100.00)	0 (0.00)
Non-evidence of disease ATA low risk (NED-L)	71 (55.04)	71 (100.00)	0 (0.00)	0 (0.00)

* *n* = 4 with pulmonary disease, *n* = 1 with bone disease.

**Table 3 cancers-15-02976-t003:** Comparisons of thyroglobulin, TSH at multiple time points and ^131^I administered activity among five groups of patients.

	ND Median (95%CI) (Min to Max)	DD Median (95%CI) (Min to Max)	R Median (95%CI) (Min to Max)	NED-I Median (95%CI) (Min to Max)	NED-L Median (95%CI) (Min to Max)	*p*
Tg_−30_ (ng/mL)	3.50 (1.62 to 5.00) (1.30 to 8.60)	28.80 (16.00 to 189.00) (16.00 to 189.00)	1.50 (0.40 to 2.93) (0.30 to 2.93)	0.47 (0.30 to 0.64) (0.09 to 1.50)	0.48 (0.30 to 0.63) (0.00 to 1.45)	<0.0001
Tg_0_ (ng/mL)	15.26 (5.92 to 38.43) (0.97 to 64.00)	159.40 (33.00 to 2095.00) (33.00 to 2095.00)	7.82 (0.98 to 149.00) (0.61 to 149.00)	1.38 (0.59 to 2.20) (0.00 to 25.31)	1.59 (0.43 to 3.08) (0.00 to 81.90)	<0.0001
Tg_0H_	33.70 (2.49 to 64.00) (2.49 to 64.00)	359.00 (159.40 to 2095.00) (159.40 to 2095.00)	74.99 (0.98 to 149.00) (0.49 to 149.00)	1.97 (0.30 to 6.54) (0.00 to 14.68)	2.73 (1.33 to 4.80) (0.00 to 39.80)	0.002
Tg_0TSH_	7.41 (0.97 to 38.43) (0.97 to 38.43)	59.24 (33.00 to 85.49) (33.00 to 85.49)	7.15 (0.61 to 20.32) (0.61 to 20.32)	0.59 (0.00 to 3.07) (0.00 to 25.31)	0.44 (0.23 to 3.08) (0.00 to 81.90)	0.004
p	0.11	0.15	0.84	0.21	0.08	
Tg_+7_ (ng/mL)	82.00 (40.00 to 308.50) (0.69 to 515.00)	169.70 (100.00 to 900.00) (100.00 to 900.00)	13.70 (1.91 to 211.20) (1.91 to 211.20)	23.81 (5.60 to 50.60) (0.00 to 99.40)	18.58 (7.39 to 37.00) (0.00 to 872.00)	0.005
Tg_+7H_	85.95 (40.00 to 515.00) (40.00 to 515.00)	534.85 (169.70 to 900.00) (169.70 to 900.00)	139.50 (88.00 to 191.00) (88.00 to 191.00)	33.85 (4.43 to 96.50) (0.00 to 99.40)	36.40 (6.39 to 78.70) (0.00 to 283.00)	0.036
Tg_+7TSH_	78.99 (0.69 to 183.00) (0.69 to 183.00)	100.00 (100.00 to 100.00) (100.00 to 100.00)	11.46 (1.91 to 211.20) (1.91 to 211.20)	18.42 (0.54 to 50.60) (0.00 to 211.20)	12.80 (5.12 to 35.00) (0.00 to 872.00)	0.37
p	0.08	0.54	0.33	0.24	0.36	
TSH_−30_ (mIU/L)	0.49 (0.13 to 0.80) (0.00 to 0.87)	0.50 (0.08 to 1.26) (0.08 to 1.26)	0.45 (0.07 to 1.35) (0.04 to 1.35)	0.11 (0.09 to 0.25) (0.00 to 0.71)	0.17 (0.12 to 0.21) (0.00 to 1.34)	0.14
TSH_0_ (mIU/L)	101.67 (79.29 to 138.38) (26.51 to 160.97)	102.86 (64.30 to 244.41) (64.30 to 244.41)	171.61 (79.34 to 498.00) (43.33 to 498.00)	106.59 (82.74 to 121.00) (36.20 to 232.43)	98.59 (84.18 to 116.10) (20.28 to 299.23)	0.32
Activity (mCi)	120 (100 to 120) (30 to 150)	150 (100 to 150) (100 to 150)	100 (80 to 150) (50 to 150)	100 (100 to 120) (50 to 120)	80 (80 to 100) (50 to 150)	<0.0001
Activity (MBq)	4440 (3700 to 4440) (1110 to 5550)	5550 (3700 to 5550) (3700 to 5550)	3700 (2960 to 5550) (1850 to 5550)	3700 (1850 to 4440) (3700 to 4440)	2960 (2960 to 3700) (1850 to 5550)	<0.0001

Abbreviations: ND = hodal disease; DD = distant disease; R = biochemical indeterminate—minimal residual tissue; NED-I = non-evidence of disease—ATA intermediate risk; NED-L = non-evidence of disease—ATA low risk Tg_−30_ = thyroglobulin 30 days before ^131^I; Tg_0_ = thyroglobulin at ^131^I; Tg_0H =_ Tg in hypothyroidism; Tg_0TSH_ = Tg after rhTSH; Tg_+7_ = thyroglobulin 7 days after ^131^I; Tg_+7H_ = Tg in hypothyroidism 7 days after ^131^I; Tg_+7TSH_ = Tg after rhTSH 7 days after ^131^I; TSH_−30_ = thyrotropin before ^131^I; TSH_0_ = thyrotropin at ^131^I. Post hoc analysis: Tg_−30_ = (ND vs. DD), *p* = 0.009; (ND vs. NED-I), *p* < 0.0001; (ND vs. NED-L), *p* < 0.0001; (DD vs. R), *p* = 0.03; (DD vs. NED-I), *p* = 0.004; (DD vs. NED-L), *p* = 0.002; Tg_0_ = (ND vs. DD) *p* = 0.03; (ND vs. NED-I) *p* = 0.0004; (ND vs. NED-L) *p* = 0.0007; (DD vs. NED-I) *p* = 0.004; (DD vs. NED-L) *p* = 0.002; Tg_0H_ = (ND vs. NED-I) *p* = 0.02; (ND vs. NED-L) *p* = 0.036; (DD vs. NED-L) *p* = 0.038; Tg_0TSH_ = (ND vs. DD) *p* = 0.0007; (DD vs. R) *p* = 0.0002; (DD vs. NED-I) *p* < 0.0001; (DD vs. NED-L) *p* < 0.0001; Tg_+7_ = (ND vs. NED-I) *p* = 0.02; (ND vs. NED-L) *p* = 0.035; (DD vs. NED-I) *p* = 0.04; Tg_+7H_ = (ND vs. DD) *p* = 0.01; (DD vs. NED-I) *p* = 0.0001; (DD vs. NED-L) *p* < 0.0001; Activity = (ND vs. NED-L) *p* = 0.001; (DD vs. NED-L) *p* = 0.007; (NED-I vs. NED-L) *p* = 0.0001.

**Table 4 cancers-15-02976-t004:** Cutoff of the thyroglobulin for detection of disease, at multiple time points.

	Nodal Disease vs. Non-Evidence Disease		Distal Disease vs. Non-Evidence Disease	
	Cutoff (Se and Sp)	AUC (95% CI)	Cutoff (Se and Sp)	AUC (95% CI)
Tg_−30_ (ng/mL)	1.30 (100 and 89.0)	99.0 (96.9 to 100)	16.00 (100.0 and 100.0)	100 (100 to 100)
Tg_0_ (ng/mL)	5.70 (85.7 and 76.8)	85.0 (75.8 to 94.6)	32.98 (100 and 96.0)	99.2 (97.4 to 100)
Tg_+7_ (ng/mL)	78.23 (76.9 and 79.0)	76.9 (63.2 to 90.6)	100.07 (100 and 87.7)	93.4 (85.3 to 100)

Abbreviations: AUC = area under curve; Se = sensitivity; Sp = specificity; %95CI = 95% confidence interval. Tg_−30_ = thyroglobulin 30 days before ^131^I; Tg_0_ = thyroglobulin at ^131^I; Tg_+7_ = thyroglobulin 7 days after ^131^I.

## Data Availability

All data are stored and available upon request to our data manager: giuseppe.campagna@uniroma1.it.

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
