# Peer review of "Predictive Role of Serum Thyroglobulin after Surgery and before Radioactive Iodine Therapy in Patients with Thyroid Carcinoma"

_cancers, 2023, doi:10.3390/cancers15112976_

Round 1

Reviewer 1 Report

This is an extremely important study showing that TG 30 days before RAI is a prognostic factor. This finding supplements other known factors and adds a new time point when re-staging of DTC patients may be considered.

 The major finding, unfortunately, is hidden among superfluous and often unnecessary information. The readers of the Journal are familiar with the basics of endocrinology and thyroid cancer; it makes possible, and necessitates, a substantial shortening of the manuscript. Major revision is required.

1.       All results have to be shown in the Results section, not elsewhere.

2.       The Manuscript has to be substantially shortened.

3.       The Discussion section should be shortened by 30 % in addition to the removal of Table 5.

Specific comments:

Line 45

“Mazzaferri et al. in 1994 demonstrated that local recurrences are frequent, especially in the cervical lymph nodes (74%), followed by thyroid remnants (20%), and then trachea or 47 muscle (6%).”

Reference is missing. The reference (1) mentioned in the previous and following sentence refers to another publication.

 Line 53

“Currently, a more individualized and risk stratified approach limits the indication for RAI treatment to high-risk patients [2].“

Ref (2) written in 1994, does not contain this current individual approach. I will not check all citations one by one; please make sure that they are appropriate and correctly numbered.

Line 89

“…disease status during follow-up (FUP) is well established…”

FUP is a rather unusual abbreviation for follow up. I suggest to spell out throughout the manuscript.

Line 127, exclusion criteria:

How Hashimoto’s thyroiditis was defined, and why was it an exclusion criterion. Please discuss it in the Discussion.

Line 242 and Table 2

These are in the Methods but would better fit the Results section.

Line 305

”All patients had no evidence of metastases at time of study.”

This is misleading. The major finding is related to metastases.

Line 400

“In our study we included only post-operative patients without AbTg and with a TSH level below 1.5 mIU/L and Tg values were measured by ultrasensitive test.“

This statement in the Discussion is not clear as nothing is said about these inclusion criteria in the Methods section, where the reported inclusion criteria are completely different.

Line 437

This part must be substantially shortened, and Table 5 removed. Now new findings (i.e. findings not mentioned in the Results section) should be shown in the Discussion.

Line 467

“…we showed that serum Tg, measured, with highly sensitive assays, 40 days after surgery, in euthyroidism, is a reliable prognostic marker…”

In the Abstract, the same is said about day 30 before RAI, and nothing is said about 40 days after surgery.

The manuscript would benefit from substantial shortening. I give some examples. Some of the following could easily be omitted:

Lines 64, the detailed description of TG, and production and function. The first half of the paragraph is common knowledge.

Line 167, the majority of the paragraph

Lines 200 to 202: May be deleted.

Lines 328 to 352: Anything shown in Table 3 need not to be repeated in the text.

Lines 355-369: The majority written here is not related to the purpose of the study and may be omitted.

The manuscript needs careful check for typing errors. E.g.:

Line 156: “of the physicians“ should read: by the physicians.

Line 285: “…the Youden’s index was those chosen for to identify the threshold.”

Questions:

Line 214, WBS was performed at 12, 18, 24 or 36 months after RAI. In patients with stimulated TG below 0.1 these WBS are not justified. Please comment.

A Figure showing the TG curves (two curves: mean TG of ND and NED patients) with time (all subsequent timepoints) could substantially add to the value of the manuscript.

Minor language and typing errors.

Author Response

Reply enclosed

Reviewer 2 Report

This work is a novel contribution to understanding the prognostic value of serum Thyroglobulin in patients with differentiated thyroid carcinoma. The authors have performed a study in patients with differentiated thyroid carcinoma, treated with thyroid surgery and 131I for remnant ablation. They found that serum Tg value (in euthyroidism 30 days before RIA) is a prognostic factor for future nodal or distal disease. The scientific and medical community will greatly benefit from this work.   Minor comments 1) It would be helpful to include more details about the collection of blood samples. Would the time of the day when samples were collected impact the TSH and Tg levels? 2) I would be helpful to know if Tg antibodies impact the levels of Tg detected by the Tg assay used to determine serum Tg levels in this work. 

Author Response

Reply enclosed

Reviewer 3 Report

Signore et al. presented a very careful study with some interesting results. They found that serum Tg measured 30 days before RAI is a reliable predictor of developing a tumor metastasis which is a very well know indicator of bad prognosis in thyroid cancer. However, the study presented some points that should be addressed.

1.       The study is only focused on Papillary Thyroid Carcinoma (PTC); using in the tittle and through the paper the terminology of Differentiated Thyroid Carcinoma is confusing since this term included Follicular Thyroid Carcinoma (FTC) as well.

2.       In the Patients and methods section the authors wrote that aggressive histological PTC variants (i.e., tall cell) were excluded but in Table 1 it appears that 5 patients with this variant were included. Please clarify.

3.       In the statistical analysis section, the authors wrote that all continuous variables are shown as mean+SD, nonetheless in table 3 (the only place in the manuscript where this could apply) the authors described the variables as medians. Additionally, in the results section the authors the authors describe that differences were found in medians, but I suspected that, as the say in Methods, they use a statistical test based on the significant difference between the means of two groups or more groups. Please clarify.

4.       Figure 2 lacks some arrows to see the flow of the medical records excluded.

5.       What are the p values placed at the left part of the table 3, what are the authors comparing? Please specify.

6.       The authors defined NED-L and NED-I, however in some point they use NED to refer to group of patients, this category has not been defined. Is it the adding of both categories?

7.       It could be interesting to calculate the ROC curve of patients with metastasis (nodal or distal) vs. NED-L/I patients.

8.       I really think that your patient’s classification is a very important part of the article in comparison to ATA guidelines. Please critically compare both.

The study is done very carefully and is indeed really detailed, the conclusions are important to improve PTC patients’ classification which is especially important in this disease.

The paper is very extensively and densely written. I suggest trying to reduce it, especially the introduction and the methodology. The main issue, however, is the results section that should be rewritten, there are phrases without verb and really strange grammatical constructs. For example: line 335 TG0 median values were differ among different groups ??

Author Response

Reply enclosed

Round 2

Reviewer 1 Report

All my comments and requests fulfilled. I suggest acceptance of the paper.

All my comments and requests fulfilled. I suggest acceptance of the paper.